

# Isolation of an antimicrobial compound produced by bacteria associated with reef-building corals

Jean-Baptiste Raina[1,2,3,4,5], Dianne Tapiolas[2], Cherie A. Motti[2], Sylvain Foret[3,6], Torsten Seemann[7], Jan Tebben[8,9], Bette L. Willis[3,4] and David G. Bourne[2,4]

[1] Climate Change Cluster (C3), University of Technology Sydney, Sydney, NSW, Australia
[2] Australian Institute of Marine Science, Townsville, QLD, Australia
[3] Australian Research Council Centre of Excellence for Coral Reef Studies, James Cook University, Townsville, QLD, Australia
[4] Marine Biology and Aquaculture, College of Science and Engineering, James Cook University of North Queensland, Townsville, QLD, Australia
[5] AIMS@JCU, James Cook University, Townsville, QLD, Australia
[6] Research School of Biology, Australian National University, Canberra, ACT, Australia
[7] Victorian Life Sciences Computation Initiative, University of Melbourne, Melbourne, Victoria, Australia
[8] Section Chemical Ecology, Alfred Wegener Institute, Bremerhaven, Germany
[9] University of New South Wales, Sydney, NSW, Australia

Corresponding author
Jean-Baptiste Raina,
Jean-Baptiste.Raina@uts.edu.au

## ABSTRACT

Bacterial communities associated with healthy corals produce antimicrobial compounds that inhibit the colonization and growth of invasive microbes and potential pathogens. To date, however, bacteria-derived antimicrobial molecules have not been identified in reef-building corals. Here, we report the isolation of an antimicrobial compound produced by *Pseudovibrio* sp. P12, a common and abundant coral-associated bacterium. This strain was capable of metabolizing dimethylsulfoniopropionate (DMSP), a sulfur molecule produced in high concentrations by reef-building corals and playing a role in structuring their bacterial communities. Bioassay-guided fractionation coupled with nuclear magnetic resonance (NMR) and mass spectrometry (MS), identified the antimicrobial as tropodithietic acid (TDA), a sulfur-containing compound likely derived from DMSP catabolism. TDA was produced in large quantities by *Pseudovibrio* sp., and prevented the growth of two previously identified coral pathogens, *Vibrio coralliilyticus* and *V. owensii*, at very low concentrations (0.5 µg/mL) in agar diffusion assays. Genome sequencing of *Pseudovibrio* sp. P12 identified gene homologs likely involved in the metabolism of DMSP and production of TDA. These results provide additional evidence for the integral role of DMSP in structuring coral-associated bacterial communities and underline the potential of these DMSP-metabolizing microbes to contribute to coral disease prevention.

## INTRODUCTION

Coral reefs are one of the most biologically diverse ecosystems on the planet (*Bellwood & Hughes, 2001*; *Knowlton, 2001*; *Pauley, 1997*). Each square centimeter of a coral's surface harbors several thousand species of microbes (*Sunagawa, Woodley & Medina, 2010*) and up to $10^7$ bacterial cells (*Garren & Azam, 2010*; *Koren & Rosenberg, 2006*), an abundance ten times greater, on average, than the surrounding sea water. These microbial assemblages are often highly specific to their coral host and include large numbers of rare and sometimes even unique taxa (*Sunagawa, Woodley & Medina, 2010*). Although the phylogenetic diversity and dynamics of coral-associated bacterial communities have been studied for more than a decade (*Bourne & Munn, 2005*; *Bourne et al., 2013*; *Littman et al., 2009*; *Rohwer et al., 2001*; *Sunagawa, Woodley & Medina, 2010*), their ecological and functional roles in the biology and health of corals are still poorly understood.

Recent studies have started to unravel the roles that coral-associated bacteria and their interactions with their coral hosts are likely to play within the coral holobiont. For example, some members of the *Cyanobacteria*, *Rhizobiales* and *Vibrionaceae* taxa are likely to fix dissolved nitrogen, a particularly important process in oligotrophic environments such as coral reefs (*Lema, Willis & Bourne, 2012*; *Lesser et al., 2004*; *Olson et al., 2009*). Others, like *Roseobacter*, *Pseudomonas* and *Oceanospirillales*, can metabolize dimethylsulfoniopropionate (DMSP), an organic sulfur compound produced in large quantities by corals and suspected to play a role in structuring coral-associated bacterial communities (*Raina et al., 2010*; *Raina et al., 2013*). It has also been hypothesized that bacteria act as a line of defense against invasive pathogens, either by competing for space and occupying coral niches (*Ritchie & Smith, 2004*), or by directly producing antimicrobial compounds that inhibit the growth of invasive microbes in coral mucus (*Ritchie, 2006*).

In artificial culture conditions, approximately 25% of the cultivable coral bacteria produce antimicrobial compounds that prevent the growth of pathogenic micro-organisms (*Ritchie, 2006*; *Shnit-Orland & Kushmaro, 2008*). Several of these antimicrobial-producing taxa, such as *Pseudoalteromonas*, *Pseudomonas*, and the *Roseobacter* clade are commonly found in association with numerous coral species (*Nissimov, Rosenberg & Munn, 2009*; *Radjasa et al., 2008*; *Rypien, Ward & Azam, 2010*; *Shnit-Orland & Kushmaro, 2009*). Although the presence of antimicrobial defences in reef-building corals has been reported (*Geffen, Ron & Rosenberg, 2009*; *Geffen & Rosenberg, 2005*; *Gochfeld & Aeby, 2008*; *Koh, 1997*), only few active compounds—all produced by the coral animal itself—have been isolated to date (*Fusetani et al., 1996*; *Kodani et al., 2013*; *Vidal-Dupiol et al., 2011*). The aim of this study was to identify specific antimicrobial compounds and thereby enhance our understanding of the functional roles played by coral-associated bacteria. Our specific objectives were to: (*i*) isolate a common coral-associated bacterium with antimicrobial activity and identify the compound(s) responsible for the activity; (*ii*) evaluate the susceptibility of the coral pathogens *Vibrio coralliilyticus* and *Vibrio owensii* to the isolated compound; (*iii*) determine the effect that

thermal stress might have on its production; and (*iv*) investigate the natural abundance of the antimicrobial compound in coral extracts.

## MATERIALS AND METHODS

### Bacterial isolation

Healthy colonies of the corals *Pocillopora damicornis*, *Acropora millepora* and *Montipora aequituberculata* (one colony per species) were collected in November 2011 from Davies Reef, Great Barrier Reef, Australia (latitude, 18°51′S; longitude, 147°41′E, Great Barrier Reef Marine Park Authority permit G12/35236.1) and maintained in aquaria for six days at the Australian Institute of Marine Science (Townsville, Queensland, Australia). Five replicate coral fragments (approximately 25 mm in length, containing 60–70 polyps) were collected from each colony and washed in sterile artificial seawater (ASW) to remove loosely attached microbes. Tissue slurries were produced by airbrushing (80 lb/in$^2$) each coral fragment into 5 mL of ASW to remove coral tissues and associated microbes. These tissue slurries were homogenized to break down tissue clumps, and a dilution series was plated immediately on bacteriological agar (1%) in 1 L ASW supplemented with 0.3% casamino acids and 0.4% glucose (*Hjelm et al., 2004*). After two days of incubation at 28 °C, single colonies were transferred into Marine Broth (MB; Difco, BD, Franklin Lakes, NJ) and grown overnight. Liquid cultures were re-plated on minimal marine agar and the procedure was repeated until pure cultures were obtained.

### Well diffusion assay with bacterial isolates

Fifty bacteria isolated from the coral tissue slurries of the three species (*A. millepora* = 16, *P. damicornis* = 17, *M. aequituberculata* = 17) were tested for growth-inhibitory activity against the known coral pathogens *Vibrio coralliilyticus* P1 (LMG23696) and *V. owensii* DY05 (LMG25443) in a well diffusion agar assay. In brief, the *Vibrio* strains were seeded into two different batches of minimal marine agar (after the agar temperature cooled to 40 °C). Following solidification, wells (diameter 5 mm) were cut into the agar and loaded with 20 µL of overnight cultures (10$^8$ cells/mL) of the test isolates grown in MB (28 °C, 170 rpm). Plates were incubated at 28 °C and monitored every 24 h for a period of 72 h for inhibition zones. *Phaeobacter* strain 27-4 was used as a positive antagonistic control on each plate because of its broad spectrum inhibitory activity against *Vibrio* (*Bruhn, Gram & Belas, 2007*; *Hjelm et al., 2004*).

### DNA extraction, gene sequencing genomic analyses

One strain, P12 isolated from *Pocillopora damicornis*, produced the strongest growth-inhibitory activity against the two target *Vibrio* strains. High molecular weight genomic DNA from P12 was extracted using a miniprep phenol-chloroform based extraction. Briefly, 5 mL of overnight liquid culture of P12 (10$^8$ cells/mL) were spun in a micro-centrifuge (10,000 rcf) for 2 min. The pellet was then resuspended in 567 µL of TE buffer, 30 µL of 10% SDS and 3 µL of 20 mg/mL proteinase K. The tube was shaken thoroughly and incubated for 1 h at 37 °C. One hundred microliters of 5 M NaCl was subsequently added and the sample thoroughly mixed before adding 80 µL of

CTAB/NaCl (10% CTAB in 0.7 M NaCl). The solution was incubated for 10 min at 65 °C, extracted with an equal volume of phenol/chloroform/isoamyl alcohol and centrifuged for 10 min (10,000 rcf). The supernatant was then extracted with an equal volume of chloroform/isoamyl alcohol and centrifuged again for 10 min. The aqueous phase was transferred to a new tube, DNA precipitated with equal volume of ice-cold isopropanol, washed with 70% ethanol and dried.

The near complete 16S rRNA gene of the strain was PCR amplified with bacterial specific primers 63F and 1387R, as outlined in *Marchesi et al. (1998)*. Amplified PCR products were visualized by electrophoresis on 1% agarose gel stained with ethidium bromide. The amplified DNA was dried in a vacuum centrifuge (Savant DNA 120) and sequenced (Macrogen, Inc., Seoul, Korea). The 16S rRNA gene sequence of isolate P12 was used for phylogenetic comparisons and Maximum Likelihood trees were constructed using the ARB software.

We produced a draft genome assembly of P12. A paired-end library was prepared using the Illumina Truseq protocol (Illumina, San Diego, CA, USA), with an insert size of 169 bp and a read size of 150 bp. The library was sequenced on an Illumina MiSeq instrument at Monash University (Melbourne, Australia). The genome was assembled with the SPAdes assembler (v2.4.0) (*Bankevich et al., 2012*) and annotated with the Prokka software (v1.5.2) (*Seemann, 2014*). The presence of the genes involved in DMSP metabolism (*dmdA, dddD, dddL, dddP, dddY, dddQ, dddW*) and TDA production (*tdaA-tdaH*) was investigated by searching for homologs of the corresponding genes using reciprocal best blast hits.

## DMSP metabolic capabilities of the isolate P12

Two different minimal media were used to examine the DMSP metabolic capabilities of P12: a modified marine ammonium salt medium (MAMS) (*Raina et al., 2009*) lacking a carbon source, and a modified basal salt medium lacking a sulfur source (*Fuse et al., 2000*) (25 g of NaCl, 0.7 g of KCl, 0.05 g of $KH_2PO_4$, 1 g of $NH_4NO_3$, 0.2 g of $MgCl_2 \cdot H_2O$, 0.02 g of $CaCl_2 \cdot 2H_2O$, 0.005 g of FeEDTA, 1 g of Tris, 5 g of sodium succinate, 1.35 g of glucose in 1 L of distilled water). DMSP was added to both media (1 mm), acting either as the sole carbon or sulfur source. Five milliliters of each culture media were inoculated in triplicate with single P12 colonies and incubated at 28 °C for six days. Negative controls containing only the basal media and DMSP were used to account for possible chemical breakdown of DMSP. Bacterial growth was assessed via optical density measurement (NanoDrop, Thermo Fisher, Waltham, MA, USA). DMSP metabolism was assessed by $^1$H Nuclear Magnetic Resonance spectroscopy (NMR). Methanol ($CH_3OH$; 40 mL) was added to each culture tube, the mixture shaken vigorously and sonicated for 10 min before being dried in vacuo using a rotary evaporator (Buchi, Flawil, Switzerland). The dried extracts were resuspended in a mixture of deuterium oxide ($D_2O$; 250 μL) and deuterated methanol ($CD_3OD$; 750 μL) (Cambridge Isotope Laboratories, Andover, MA, USA). A 750-mL aliquot of the particulate-free extract was transferred into a 5-mm Norell tube (Norell Inc., Landisville, NJ, USA) and analyzed immediately using quantitative NMR (*Tapiolas et al., 2013*).

## Preparation of crude extracts for antagonist assays

An overnight culture of P12 (8 mL) was used to inoculate $4 \times 250$ mL of MB (total culture volume = 1 L). Bacterial cells were incubated for two days at 28 °C (120 rpm); the culture broth was then acidified to pH 2 with sulphuric acid before being exhaustively extracted with ethyl acetate ($3 \times 1.5$ L). The extract was washed three times with MilliQ $H_2O$ and dried in vacuo using a rotary evaporator (Buchi). The dried extract was then weighed and resuspended in $CH_3OH$ (which was chosen for its ability to solubilize a wide range of compounds, its volatility and its innocuity in small volume towards both *V. coralliilyticus* and *V. owensii*) and tested in well-diffusion assays to confirm the extraction of the antimicrobial compound(s).

## Purification and characterization of active compound

Purification of the crude extract was carried out using solid phase extraction on a reversed phase $C_{18}$ flash vacuum column (Septra $C_{18}$-E, Phenomenex, Torrance, CA, USA). Eleven fractions were eluted sequentially with 20, 40, 60, 80 and 90% $CH_3OH$ in $H_2O$ and 100% $CH_3OH$, followed by 20, 50 and 100% dichloromethane ($CH_2Cl_2$) in $CH_3OH$, 40% hexane in $CH_2Cl_2$ and finally 100% hexane. Each fraction was dried and resuspended in $CH_3OH$ (1 mg mL$^{-1}$). Well diffusion assays were prepared as described above. On each plate, test wells were inoculated with 20 µL of each chromatographic fraction, or 20 µL of $CH_3OH$ as a control, and *Vibrio* growth monitored. The most active faction (80% $CH_3OH$) presented an intense yellow color. Fine orange-red needles were crystallized from this active fraction to yield compound **1** (2.1 mg, 1.7% dry weight of organic extract).

## NMR and FTMS analysis

Identification and structural elucidation of compound **1** was achieved using liquid chromatography–mass spectrometry (LC-MS), NMR, and Fourier Transform mass spectrometry (FTMS). Likewise these techniques were used to monitor for the presence of compound **1** in extracts and fractions. LC-MS analyses were performed on a Thermo Fisher Scientific Ultra High Performance Liquid Chromatography system connected to an LTQ Orbitrap XL mass spectrometer (Thermo Fisher Scientific, San Jose, CA, USA). Samples were separated on a ACQUITY UPLC BEH RP-$C_{18}$ column (130 Å, 1.7 µM, $2.1 \times 100$ mm, solvents A = aqueous 0.1% formic acid and B = acetonitrile, gradient elution 80% A: 20% B for 0.5 min ramped up to 100% B over 10 min, then held for 4 min, 400 µL) and detected by positive mode electrospray ionisation using two different *m/z* ranges: 150–1,500 and 170–400. $^1$H and $^{13}$C NMR spectra of compound **1** were acquired in a 5 mm 509-UP Norell NMR tube on a Bruker Avance 600 MHz NMR spectrometer (Bruker, Germany) with a TXI cryoprobe using standard Bruker pulse sequences. NMR spectra were referenced to residual $^1$H and $^{13}$C resonances in deuterated chloroform ($CDCl_3$). High resolution mass spectra of compound **1** were measured with a Bruker BioApex 47e FTMS fitted with an Analytica of Branford ESI source; ions were detected in negative mode within a mass range *m/z* 200–1,000 via direct infusion at 120 µL h$^{-1}$.

## Temperature-dependent activity

The antimicrobial activity of P12 grown at 32 °C (upper limit of coral thermal tolerance) was compared to that of the control incubated at 28 °C. The two cultures were grown overnight in MB at the two different temperatures, and their densities were determined by flow-cytometry (BD Accuri C6, Beckman Coulter, Brea, CA, USA). Cell numbers were normalized prior to inoculation into agar wells, and their activities against the two pathogens were compared using well-diffusion assays as described above. The same procedure was repeated with compound **1**: two vials containing equal concentrations (2 $\mu$M of **1** in $CH_3OH$) were incubated overnight at 28 or 32 °C and their antimicrobial activities compared using the well diffusion assay.

## Preparation of coral extracts

The coral species *Montipora aequituberculata, M. turtlensis, Pocillopora damicornis, Acropora millepora,* and *Porites cylindrica* (one colony each; 500 g of dry skeleton per species) were collected in July 2012 from Orpheus Island, Great Barrier Reef, Australia (latitude, 18°35′S; longitude, 146°20′E, Great Barrier Reef Marine Park Authority permit G12/35236.1). Coral tissues were airbrushed (80 lb/in$^2$) into 0.2 $\mu$M filtered seawater (FSW) (total volume = 500 mL), acidified to pH 2 with sulphuric acid and the solution exhaustively extracted with equal volumes of ethyl acetate (3 × 750 mL). The combined organic layers were partitioned with MilliQ $H_2O$, dried and tested in well-diffusion assays, as previously described for the bacterial isolate extracts. The extracts of those coral species that exhibited antimicrobial activity were subsequently fractionated as described above for the crude extract from P12 and tested in well-diffusion assays. The active fractions were analyzed using $^1$H NMR, FTMS and LC-MS.

## RESULTS

### Isolate P12: antimicrobial production, taxonomy and metabolic capabilities

A total of 50 coral-associated bacterial isolates were obtained from tissue slurry homogenates of the three coral species. Twelve of the 50 strains tested against the two pathogenic *Vibrios* (*V. coralliilyticus* and *V. owensii*) inhibited their growth in well diffusion assays. The bioactive isolate that exhibited the strongest in vitro activity against both pathogens, isolate P12, originated from *Pocillopora damicornis* and produced growth inhibition zones of 5 mm (±0.07 mm, $n = 20$) against *V. owensii* and 2 mm (±0.09 mm, $n = 20$) against *V. coralliilyticus*. The activity of P12 was temperature-dependent (Figs. 1A and 1B) and was significantly reduced when grown at 32 °C compared to 28 °C (Unpaired T-Test, $n = 20$, df = 38, $t = 30.61$, $^*p < 0.001$ for *V. owensii* and $n = 20$, df = 38, $t = 10.49$, $^*p < 0.001$ for *V. coralliilyticus*; Fig. 1C). Based on its bioactivity, the isolate P12 was selected for bioassay-guided fractionation.

According to its 16S rRNA gene sequence (NCBI accession number: KX198136), isolate P12 is an alphaproteobacterium belonging to the *Rhodobacteraceae* family and the *Pseudovibrio* genus. Its most closely related species is *Pseudovibrio denitrificans* (100% identity to the type strain; Fig. 2). Like other *P. denitrificans* strains

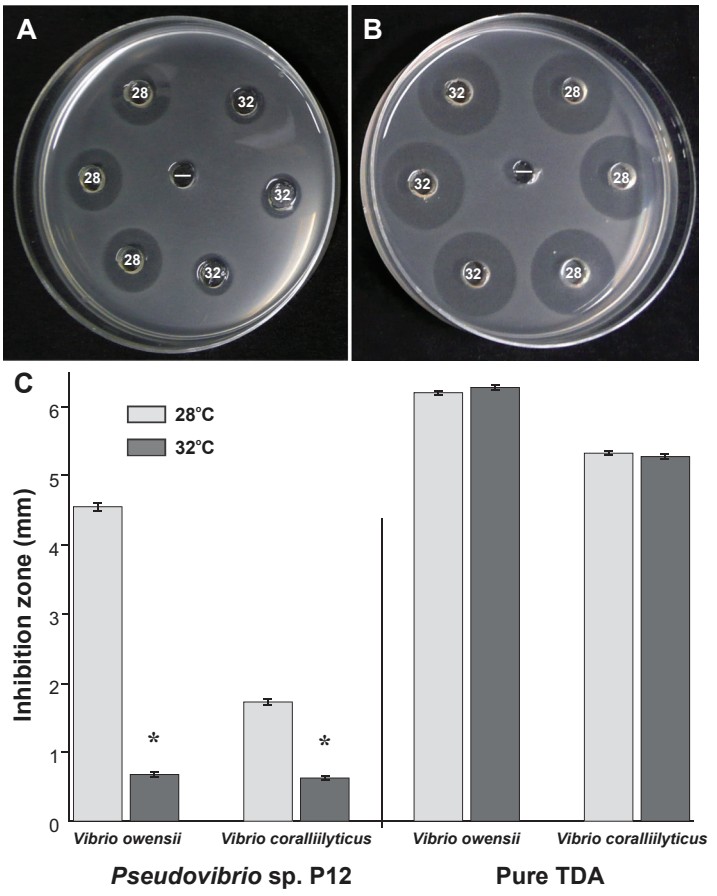

**Figure 1 Representative well diffusion assays of (A) *Pseudovibrio* sp. P12 and (B) pure TDA, incubated at two different temperatures (28 and 32 °C) and then inoculated onto agar plates with embedded *Vibrio owensii* ((−) Negative control).** (C) Comparison of the radius of inhibition zones between the two temperature treatments for both *Pseudovibrio* sp. P12 (Unpaired T-Test, $n = 20$, df = 38, $t = 30.61$, *$p < 0.001$ for *V. owensii* and $n = 20$, df = 38, $t = 10.49$, *$p < 0.001$ for *V. coralliilyticus*) and pure TDA (2 μM, Unpaired T-Test, $n = 20$, df = 38, $t = −0.94$, $p = 0.355$ for *V. owensii* and $n = 20$, df = 38, $t = 0.632$, $p = 0.531$ for *V. coralliilyticus*).

(*Enticknap et al., 2006*), P12 colonies formed brown mucoid colonies when grown on Marine Agar. The brown color was absent when the strain was grown on minimal marine agar, with colonies appearing white. This strain was able use DMSP as either a sole carbon or sole sulfur source (Fig. 3). The complete utilization of DMSP from the liquid media after 2–3 days of incubation, as well as the presence of its metabolic byproduct dimethylsulfide (DMS), were confirmed by ¹H NMR. However acrylate, another possible byproduct of DMSP metabolism, was not observed.

Among the seven different DMSP degradation pathways currently identified (*Moran et al., 2012*), the full DMSP cleavage pathway (*dddD*, *dddB*, *dddC*, *dddT*, *dddR*; Table 1), involved in the conversion of DMSP into DMS without formation of acrylate (*Todd et al., 2007*) (Table 1), was identified in P12. We also identified possible orthologs for the demethylation pathway (*dmdA*, *dmdB*, *dmdC* and *dmdD*) used by marine bacteria to assimilate sulfur from DMSP, though these gene have low sequence identity to the genes

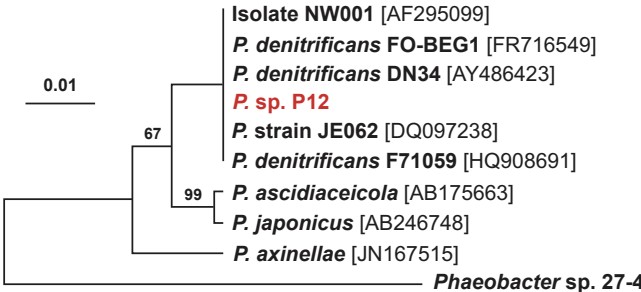

**Figure 2 Maximum likelihood phylogenetic tree based on 16S rRNA gene sequences showing the isolate used in this study (P12 in red) and closely associated *Pseudovibrio* spp.** Note: the strain FO-BEG1 has been fully sequenced. *Phaeobacter* sp. 27-4 (AJ536669) was used as outgroup. Maximum parsimony bootstrap values (10,000 replicates) are given when different from 100. The scale bar indicates the number of substitution per nucleotide position.

originally identified in *Ruegeria pomeroyi* DSS-3 (*Howard et al., 2006*; *Reisch et al., 2011*) (Table 1). The presence of these two gene pathways corroborates the $^1$H NMR measurements: the observed production of DMS without acrylate formation following DMSP metabolism (DddD pathway); and the ability to use DMSP as sole sulfur source (DmdA pathway) (Table 1).

## Identification of antimicrobial compounds produced by P12

Well diffusion assays revealed that the crude extract from P12 retained the antimicrobial properties of the strain against both *Vibrio* species. Purification of the active fractions using reverse phase liquid chromatography yielded compound **1**: optically inactive orange-red crystals; 2.1 mg (1.7% dry weight); IR (film) $\nu_{max}$ 3,420, 1,660, 1,280 cm$^{-1}$; UV (PDA, CH$_3$OH) $\lambda_{max}$ 512 nm; $^1$H NMR spectrum (600 MHz, CD$_3$Cl): $\delta$ 7.12, 7.44, 7.45 and 16.7; $^{13}$C NMR (150 MHz, CD$_3$Cl): $\delta$ 120.3, 132.0, 136.0, 138.7, 149.5, 168.7, 171.7, and 183.5; HRESIMS $m/z$ found 210.9534 (calculated for C$_8$H$_3$O$_3$S$_2^-$ 210.9529, $\Delta$ 2 ppm). Combined spectroscopic techniques revealed that compound **1** was tropodithietic acid (TDA) (*Brinkhoff et al., 2004*; *Penesyan et al., 2011*) (Fig. 4A).

Orthologs for 11 genes involved in TDA biosynthesis (*Geng et al., 2008*) were present in the *Pseudovibrio* sp. P12 genome (Table 1). The biosynthesis of TDA correlated with production of the yellow-brown pigmentation in the culture medium and antimicrobial activity, similar to that previously reported (*Brinkhoff et al., 2004*; *Bruhn et al., 2005*; *Porsby, 2010*). Both coral pathogens were highly sensitive to TDA, with the pure compound still visually inhibiting their growth at 0.5 µg/mL (2.35 µM; Fig. 4B). In contrast to the decrease in antimicrobial activity exhibited by *Pseudovibrio* sp. P12 after incubation at 32 °C, TDA activity was not affected by exposure to this temperature (Unpaired T-Test, $n = 20$, df = 38, $t = -0.94$, $p = 0.355$ for *V. owensii* and $n = 20$, df = 38, $t = 0.632$, $p = 0.531$ for *V. coralliilyticus*; Figs. 1B and 1C).

## Investigating the presence of TDA in coral samples

All of the extracts derived from the coral species investigated exhibited antimicrobial activity against the two pathogens, with the inhibition zones for *P. cylindrica*,

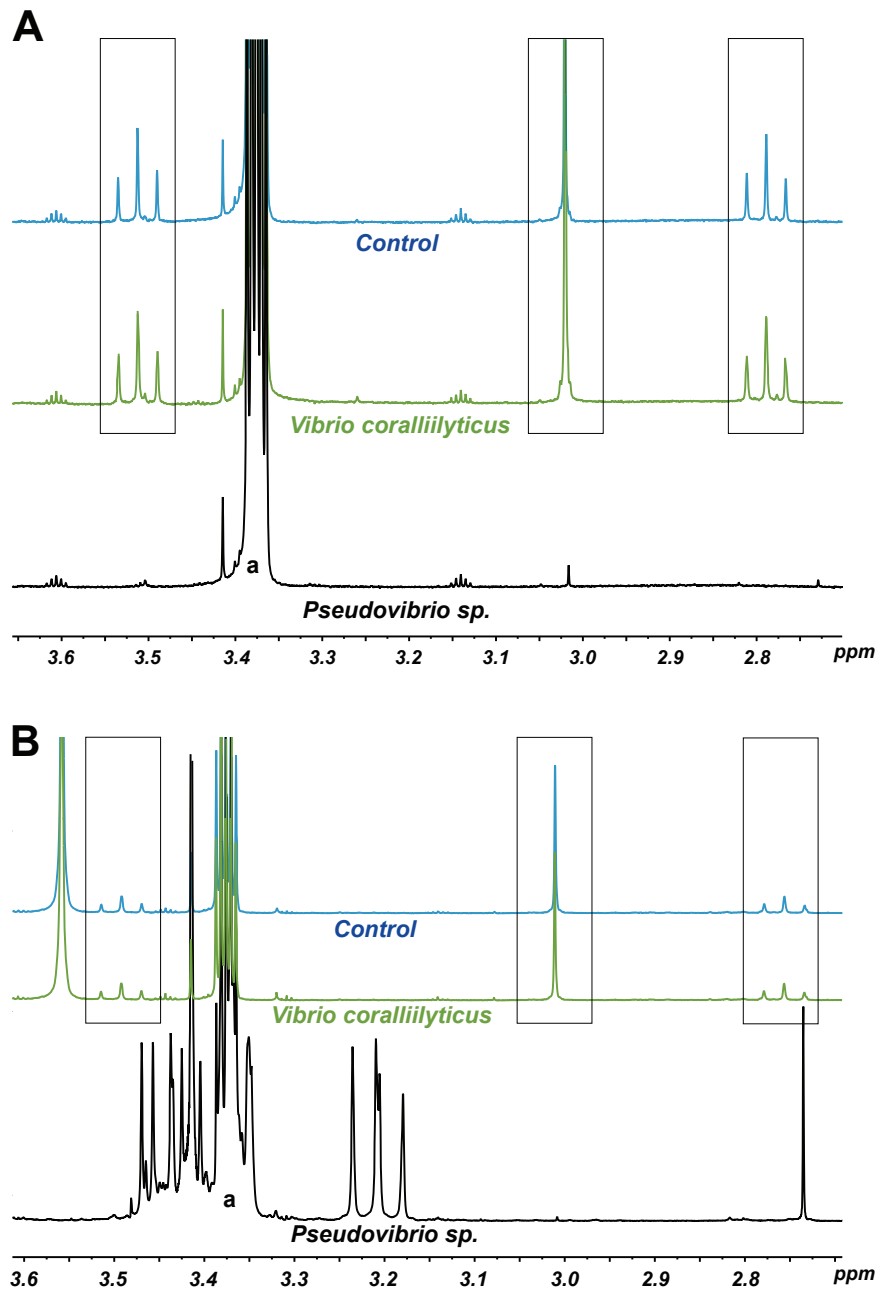

**Figure 3** **¹H NMR spectra showing DMSP utilization as (A) the sole carbon source and (B) the sole sulfur source in minimal media at the end of a six-day incubation.** The "control" lines in all cases are the growth medium (with no bacterial inoculation). The black and green spectra show the results from inoculation with *Pseudovibrio* sp. P12 and *V. coralliilyticus* (negative control), respectively. In both cases, the DMSP signals (within the three boxes, see *Tapiolas et al. (2013)*) disappeared in the *Pseudovibrio* treatment and remain unchanged between the no-bacteria control and the *V. coralliilyticus* treatment. In the case of DMSP as a sole sulfur source, *Pseudovibrio* consumed the DMSP and other carbon sources present and produced secondary metabolites (appearance of new signals). a: solvent peak (methanol).

*M. aequituberculata*, *M. turtlensis* and *P. damicornis* ranging from 3–5 mm in radius whilst the inhibition zones for *A. millepora* were much smaller (1 mm on average). ¹H NMR, LC-MS and FTMS analyses of the extracts and the active fractions of all coral

**Table 1 Orthologous genes involved in DMSP degradation and TDA biosynthesis in *Pseudovibrio* sp. P12 genome.** Accession numbers available in NCBI (http://www.ncbi.nlm.nih.gov/genbank/).

| Gene | Function | Percent of identity (%) | Accession number |
|------|----------|-------------------------|------------------|
| *DddD* | L-carnitine dehydratase | 70[a] | KM819464 |
| *DddT* | BCCT transporter | 70[a] | KM819465 |
| *DddB* | Alcohol dehydrogenase | 70[a] | KM819466 |
| *DddC* | Methylmalonate-semialdehyde dehydrogenase | 56[a] | KM819467 |
| *DddR* | Transcriptional regulator, LysR family protein | 53[a] | KM819468 |
| *DmdA* | Aminomethyl transferase family protein | 24[b] | KU521525 |
| *DmdB* | Acyl-CoA synthetase | 30[b] | KU521526 |
| *DmdC* | MMPA-CoA dehydrogenase | 43[b] | KU521527 |
| *DmdD* | MTA-CoA hydratase | 32[b] | KU521528 |
| *Alma1* | Hypothetical Alma1 ortholog | 99[c] | KU521524 |
| *tdaA* | Transcriptional regulator, LysR family protein | 66[d] | KU760700 |
| *tdaB* | Glutathione S-transferase domain protein | 67[d] | KU760701 |
| *tdaC* | Prephenate dehydratase | 67[d] | KU760702 |
| *tdaD* | Acyl-CoA thioester hydrolase | 81[d] | KU760703 |
| *tdaE* | Acyl-CoA dehydrogenase | 69[d] | KU760704 |
| *tdaF* | Phosphopantothenoylcysteine decarboxylase | 72[d] | KU760705 |
| *cysI* | Sulfite reductase hemoprotein beta-component | 76[e] | KU760706 |
| *malY* | Cystathione beta-lyase | 68[e] | KU760707 |
| *paaI* | Phenylacetate-CoA oxygenase | 59[f] | KU760708 |
| *paaJ* | Phenylacetate-CoA oxygenase, PaaJ subunit | 73[g] | KU760709 |
| *paaK* | Phenylacetic acid degradation oxidoreductase | 57[g] | KU760710 |

**Notes:**
Percent of similarity compared to:
[a] *Marinomonas* sp. MWYl1 (*Todd et al., 2007*);
[b] *Ruegeria pomeroyi* DSS-3 (*Reisch et al., 2011*);
[c] *Pseudovibrio* sp. FO-BEG1 (*Alcolombri et al., 2015*);
[d] *Phaeobacter inhibens* (*Brock, Nikolay & Dickschat, 2014*);
[e] *Nesiotobacter exalbescens*;
[f] *Rhodopseudomonas palustris*;
[g] *Stappia stellulata*.

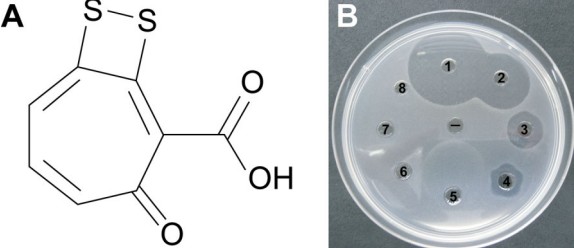

**Figure 4 (A) Tropodithietic acid (TDA).** (B) Dilution series of pure TDA, showing zones of growth inhibition against *Vibrio coralliilyticus*: ((1) 500 µg/mL, (2) 50 µg/mL, (3) 5 µg/mL, (4) 500 ng/mL, (5) 50 ng/mL, (6) 5 ng/mL, (7) 500 pg/mL, (8) 50 pg/mL, (−) negative control with solvent only).

species did not confirm the presence of TDA. The purified TDA could be detected by LC-MS in femtomolar concentrations when the coral fractions were artificially spiked, indicating that this lack of detection was not due to preferential ionization. Thus, TDA was

either not present in the coral fractions tested or in concentrations below the LC-MS detection threshold.

## DISCUSSION

While previous studies have identified corals as a source of bacteria with antimicrobial activity (*Radjasa et al., 2008*; *Sulistiyani et al., 2010*), this study represents the first isolation and identification of an antimicrobial compound produced by a reef building coral-associated bacterium with activity against coral-associated pathogens. TDA was characterized from a pure culture of *Pseudovibrio* sp. P12 previously isolated from corals and strongly inhibited the growth of two coral pathogens. Although we could not confirm the presence of this molecule in the corals tested, TDA has the capacity to provide protective antimicrobial properties to the coral host and prevent colonization by invasive bacterial species.

The strain P12 strongly inhibited the growth of *Vibrio coralliilyticus* and *V. owensii*, two coral pathogens causing white syndrome (a collective term describing rapidly progressing tissue loss, exposing band-like areas of white skeleton) (*Ben-Haim et al., 2003*; *Sussman et al., 2008*; *Ushijima et al., 2012*; *Willis, Page & Dinsdale, 2004*). *Vibrio coralliilyticus* exhibits antimicrobial resistance to a wide range of commercial antibiotics and is also resistant to the activities of a large number of coral-associated bacteria (*Rypien, Ward & Azam, 2010*; *Shnit-Orland & Kushmaro, 2009*; *Vizcaino et al., 2010*). Its resistance to commercial antibiotics is considerably greater than that of other marine pathogens such as *V. parahaemolyticus* or *V. vulnificus*, and may contribute to its competitive advantage within the coral holobiont, as well as its ability to infect corals (*Vizcaino et al., 2010*). However, whilst *V. coralliilyticus* is resistant to many coral-associated bacteria, its growth was strongly inhibited by the strain P12, emphasizing its antimicrobial capabilities.

The isolate P12 belongs to the bacterial genus *Pseudovibrio* (*Shieh, Lin & Jean, 2004*), and is ubiquitously found in association with healthy sponges (*Enticknap et al., 2006*; *Thiel & Imhoff, 2003*; *Webster & Hill, 2001*) and corals (see Table 2). *Pseudovibrio* are thought to be involved in symbiotic relationships with various organisms; they are vertically transmitted in large densities by adult sponges to their larvae (*Enticknap et al., 2006*) and their presence is required for the growth of the sulfur-oxidizing bacteria *Beggiatoa* in culture (*Bondarev et al., 2013*). Furthermore, their genome is organized similarly to that of *Rhizobia*, a well-characterized symbiotic bacterium (*Bondarev et al., 2013*; *Enticknap et al., 2006*; *Kennedy et al., 2009*). The full genome sequences of *Pseudovibrio* FO-BEG1 (KEGG genome T01669; isolated from a Caribbean coral, and sharing 100% sequence similarity with P12 based on its 16S rRNA gene sequence) reveal the presence of genes involved in host-cell adhesion, interactions with eukaryotic cell machinery, and production of secondary metabolites (*Bondarev et al., 2013*), further suggesting that this bacterium is involved in symbiotic relationships with its hosts.

The *Pseudovibrio* genus is also known for its antimicrobial properties, especially against human pathogens such as *Mycobacterium tuberculosis*, *Bacillus cereus*, *Yersinia*

**Table 2 Summary of *Pseudovibrio* isolated or sequenced from corals, accession numbers are displayed when available.**

| Host | Location | Method | Reference | Accession numbers |
|---|---|---|---|---|
| *Acropora palmata* | Panama | Amplicon | *Sunagawa, Woodley & Medina (2010)* | GU118050, GU118108, GU119014 |
| *Porites astreoides* | Panama | Amplicon | *Sunagawa, Woodley & Medina (2010)* | GU118050, GU118108, GU119014 |
| *Acropora cervicornis* | Panama | Amplicon | *Sunagawa, Woodley & Medina (2010)* | GU118050, GU118108, GU119014 |
| *Montastrea franksi* | Panama | Amplicon | *Sunagawa, Woodley & Medina (2010)* | GU118050, GU118108, GU119014 |
| *Tubastraea coccinea* | China | Amplicon | *Yang et al. (2013)* | JF925014 |
| *Pseudopterogorgia americana* | Puerto Rico | Isolated | *Vizcaino et al. (2010)* | GQ406787, GQ406798, GQ391966, GQ406786 |
| *Platygyra carnosus* | Hong Kong | Isolated | *Chiu et al. (2012)* | JF411474, JF411466, JF411439, JF411464 |
| *Oculina patagonica* | Israel | Isolated | *Koren & Rosenberg (2006)* | DQ416557, AY654776 |
| *Montastrea anularis* | Florida | Isolated | *Rypien, Ward & Azam (2010)* | FJ952798, FJ952774, FJ952804 |
| *Sinularia* sp. | Indonesia | Isolated | *Sulistiyani et al. (2010)* | NA |
| *Acropora almata* | Florida | Isolated | *Ritchie (2006)* | DQ530540 |
| *Sarcophyton* sp. | Java | Isolated | *Sabdono & Radjasa (2006)* | NA |
| *Oculina patagonica* | Israel | Isolated | *Nissimov, Rosenberg & Munn (2009)* | NA |
| *Lobophytum* sp. | Taiwan | Isolated | *Chen et al. (2012)* | JQ342682, JQ342695, JQ342696, JQ342697 |
| Hard coral | Florida | Isolated | *Bondarev et al. (2013)* | CP003147 |

**Notes:**

NA, not available.

enterocolitica, *Listeria monocytogenes* or methicillin-resistant *Staphylococcus aureus* (*O'Halloran et al., 2011*; *Sulistiyani et al., 2010*). To date, three active compounds have been isolated from different *Pseudovibrio* strains: heptylprodigiocin in tunicate-associated *P. denitrificans* Z143-1 (*Sertan-de Guzman et al., 2007*), pseudovibrocin in *P. denitrificans* strain PaH3.28 (*Vizcaino, 2011*) and TDA from red alga-associated *P. ascidiaceicola* D323 (*Penesyan et al., 2011*). In the present study, we isolated TDA from P12, a strain closely related to *P. denitrificans*. The ability of P12 to produce TDA was further supported by the the presence of the genes essential for TDA biosynthesis in the genome of strain P12 (*Geng et al., 2008*). TDA inhibits the growth of a wide range of marine pathogens (*Bruhn, Gram & Belas, 2007*; *Bruhn et al., 2005*) and is produced almost exclusively by bacteria from the *Roseobacter* clade, especially the genera *Phaeobacter, Silicibacter*, and *Ruegeria* (*Brinkhoff et al., 2004*; *Bruhn et al., 2005*; *Geng & Belas, 2010*; *Geng et al., 2008*; *Wilson et al., 2016*) that are commonly associated with DMSP-producing dinoflagellates (*Miller et al., 2004*; *Wagner-Döbler & Biebl, 2006*) and reef-building corals (*Bourne et al., 2013*; *Littman et al., 2009*; *Raina et al., 2009*).

Many members of the *Roseobacter* clade, including coral-associated isolates, have been implicated in sulfur cycling (*Miller et al., 2004*; *Moran, González & Kiene, 2003*; *Raina et al., 2010*). Interestingly, TDA contains two sulfur atoms ($C_8H_4O_3S_2$) and *Pseudovibrio* sp. P12 was able to use DMSP either as sole carbon or sole sulfur source, a common trait among *Alphaproteobacteria* and especially the *Roseobacter* clade (*Bruhn et al., 2005*; *Wagner-Döbler & Biebl, 2006*). Bacteria from this clade preferentially metabolize DMSP rather than sulphate ($SO_4^{2-}$), despite the latter being between $10^6$–$10^7$-fold more abundant in seawater (*Geng & Belas, 2010*; *Kiene et al., 1999*). Based on genomic and chemical analyses, DMSP metabolism in P12 can occur via two likely routes: the cleavage

pathway (encoded by *dddD*, (*Todd et al., 2007*)) that releases the climate-regulating molecule DMS, and the demethylation pathway (encoded by *dmdA*, (*Howard et al., 2006*)) by which the bacterium can retain the sulfur contained in DMSP molecules.

The biosynthetic pathway of TDA has not been fully elucidated (*Brock, Nikolay & Dickschat, 2014*). Both labelling (*Cane, Wu & Van Epp, 1992*; *Thiel et al., 2010*) and genetic dissection (*Geng & Belas, 2010*) studies have shown that its aromatic skeleton is derived from phenylacetyl-CoA produced by the shikimate pathway. However, the sulfur donor allowing the incorporation of the two sulfur atoms into the TDA molecule has not been clearly identified. It has been proposed that sulfur originating from DMSP metabolism might be used to synthesize TDA (*Bruhn, Gram & Belas, 2007*; *Bruhn et al., 2005*; *Geng & Belas, 2010*; *Porsby, 2010*; *Wagner-Döbler & Biebl, 2006*). For example, DMSP increases TDA synthesis two-fold in comparison to other sulfur sources (*Geng & Belas, 2010*), suggesting that DMSP is a preferred source of sulfur for TDA biosynthesis. Even though other sources of sulfur, such as the amino-acids cysteine and methionine present in artificial media like Marine Broth, might be used to synthesize TDA (*Geng & Belas, 2010*), DMSP is by far the most readily available reduced sulfur source in the marine environment (*Simó, 2001*). It is therefore likely that in DMSP-rich environments, such as reef-building corals, DMSP metabolism provides the sulfur needed to produce TDA via the demethylation pathway (*Howard et al., 2006*).

The presence of TDA in extracts derived from five coral species (*Montipora aequituberculata*, *M. turtlensis*, *Pocillopora damicornis*, and *Porites cylindrica*) could not be confirmed. Three possibilities can explain this lack of detection: (*i*) TDA is not synthesized in corals; (*ii*) TDA is present in corals in concentrations below the detection limit of our instruments (which would imply that this compound has a very limited role in coral defense); and (*iii*) our sampling effort was not sufficient. Indeed, we only sampled one colony per species, from a location more than 100 km away from the site where the TDA-producing bacteria was isolated and without prior characterization of the bacterial communities present in the colony sampled. Given that TDA-producing *Roseobacters* are among the first bacteria to colonize the surface of marine microalgae and corals (*Apprill et al., 2009*; *Dang & Lovell, 2000*; *Miller et al., 2004*) and can be highly abundant in some coral species (*Raina et al., 2009*), it would be premature to rule out possible TDA biosynthesis in corals.

The activity of P12 against *V. coralliilyticus* sharply decreased at elevated temperatures (32 °C), however, the activity of the purified TDA did not. This reveals that the loss of antimicrobial activity observed for P12 at 32 °C is not due to thermal sensitivity of TDA but likely to a decrease in its production. Our results are in line with previously reported decline in the antibacterial activity of other TDA-producing *Roseobacter* with temperature increase (*Bruhn et al., 2005*). Clear links have previously been identified between warm thermal anomalies and outbreaks of white syndromes (*Bruno et al., 2007*; *Heron et al., 2010*; *Maynard et al., 2011*). If indeed TDA is synthesized in vivo, a decrease in its production during anomalously high seawater temperatures could facilitate pathogen outbreaks in corals following thermal stress,

especially since the virulence of some disease-causing bacteria (i.e. *V. coralliilyticus*) increase at 32 °C (*Sussman et al., 2008*).

This study demonstrates that a common coral-associated bacterium, *Pseudovibrio* sp. P12, produces TDA, a potent antimicrobial compound that inhibits the growth of marine and coral pathogens, including *V. coralliilyticus*. The bacterium can use DMSP as a sole sulfur or carbon source and potentially as a precursor in the biosynthesis of TDA. The production of TDA by *Pseudovibrio* sp. P12 is greatly reduced at temperatures causing thermal stress in corals, potentially providing a window of opportunity for the growth of pathogens. These results provide additional evidence for the integral role of DMSP in structuring healthy, coral-associated bacterial communities and suggest that these DMSP-metabolizing communities may contribute to the prevention of coral diseases.

## ACKNOWLEDGEMENTS

The authors would like to thank E. Botté, C. Gao and M. Garren for their laboratory assistance.

### Funding

This work was supported by the ARC Centre of Excellence for Coral Reef Studies, AIMS@JCU, James Cook University and the Australian Institute of Marine Science. The funders had no role in study design, data collection and analysis, decision to publish, or preparation of the manuscript.

### Competing Interests

The authors declare that they have no competing interests.

### Author Contributions

- Jean-Baptiste Raina conceived and designed the experiments, performed the experiments, analyzed the data, wrote the paper, prepared figures and/or tables, reviewed drafts of the paper.
- Dianne Tapiolas analyzed the data, reviewed drafts of the paper.
- Cherie A. Motti analyzed the data, contributed reagents/materials/analysis tools, reviewed drafts of the paper.
- Sylvain Foret analyzed the data, reviewed drafts of the paper.
- Torsten Seemann analyzed the data, reviewed drafts of the paper.
- Jan Tebben analyzed the data, reviewed drafts of the paper.
- Bette L. Willis conceived and designed the experiments, contributed reagents/materials/analysis tools, reviewed drafts of the paper.
- David G. Bourne conceived and designed the experiments, contributed reagents/materials/analysis tools, reviewed drafts of the paper.

## Field Study Permissions

The following information was supplied relating to field study approvals (i.e., approving body and any reference numbers):

Great Barrier Reef Marine Park Authority, Permit number: G12/35236.1.

## DNA Deposition

The following information was supplied regarding the deposition of DNA sequences:

GenBank: All accession numbers are provided in the main text.

## Supplemental Information

Supplemental information for this article can be found online at http://dx.doi.org/10.7717/peerj.2275#supplemental-information.

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
