# Peer review of "Isolation of an antimicrobial compound produced by bacteria associated with reef-building corals"

_PeerJ, doi:10.7717/peerj.2275_

## Round 0.1 · original submission · Minor Revisions

The reviewers found the study of great interest and considered that the manuscript is well written. However, they have suggested changes in the organization of figures and plates to help the flow of the discussion. Also, there have been a few other minor points raised by the reviewers and I recommend the authors to address them all.

Reviewer 1 ·

Basic reporting

The manuscript is well written and easy to understand.

Specific comments:

You say that you assembled a whole genome for P12. Do you have an accession number for the genome?

Line 185: should be "reversed phase"

Line 192: should be "(80% CH3OH in H2O)"

Line 366: should be "could occur"

In Figure 1: Please include the concentration of TDA in the figure legend.

In Figure 2: You don't explain what 6-8 are in Figure 2B. Either include an explanation in the legend or remove the numbers from the figure.

It would be nice to see accession numbers in Table 2.

Experimental design

The experimental design is sound.

Validity of the findings

The biggest downside to this study is that TDA was not detected in the corals themselves. Can you discuss this a bit more in your discussion and explain why you think TDA may be important in corals even though you don't see it in the coral extracts? On line 389 you say that the concentration of TDA produced in the environment is likely close to the detection limits of the instrumentation used. Can you elaborate on that? What are the detection limits? How do those limits compare to the lowest active concentration of TDA?

Can you discuss how the production of TDA and the antibacterial activity of P12 might be different in culture compared to within the coral host? Also how this relates to changes in temperature. For example, changes in temperature will undoubtedly cause many metabolic changes within the coral holobiont that could impact the growth of P12 and/or the production of TDA.

The antibacterial activity of P12 was reduced under high temperature. You assume this means that less TDA is being produced (which is probably true), but there could be other metabolic reasons for the change. Did you quantify the amount of TDA produced under both conditions? I think that would be a good addition to the manuscript if possible, but not absolutely necessary for publication. If you don't include data on the amount of TDA produced under different temperatures, you should acknowledge that although the change in activity is likely due to a decrease in TDA production, there could be other things happening as well.

Additional comments

Overall this is an excellent manuscript. It is well written and the results are clear. It is an interesting study and contributes nicely to our understanding of the complex roles of bacteria within the coral holobiont.

Reviewer 2 ·

Basic reporting

The paper is well written but the figure plates need to be reorganized to conform with when they are discussed in the text.

Experimental design

Good - but could have used one more medium (with no C or S source) - see comments to authors.

Validity of the findings

No comments.

Additional comments

This is a very interesting and, for the most part, well written paper that will be a nice addition to the field. The only problem is the organization of the figures, and plates within figures, and their sequential discussion in the text, which needs to be changed for continuity. I have made specific recommendations below. Also, it's too bad that they did not also use a medium with no C or S source, but the data are still worth publishing. Others (or the same team) can follow this up. My other comments are minor.

Line 54 - Run on sentence. Start a new sentence with 'each'.
Line 78 - change "cultivable coral-bacteria consortia" to "cultivable coral bacteria".
Line 86 says: "The aim of this study was to determine whether bacteria produce antimicrobial compounds in corals". This has already been shown. See, for example, Gantar et al., 2011, Antibacterial activity of marine and black band disease cyanobacteria against coral-associated bacteria. Marine Drugs. 9:2089-2105. There are others - look in the literature - and I later saw that two are cited in the discussion. Change the aim to 'determination and identifation of specific compounds', which is new.
Line 100 - be more specific than "less than a week".
Line 106 - this is not a minimal medium. Change to "plated immediately on bacteriological agar (1%) in 1 L ASW supplemented with 0.3% casamino acids and 0.4% glucose".
Line 118 - instead of saying "dense" cultures provide the OD or cells per ml.
Line 128 - what does "saturated liquid culture" mean? Again, provide OD or cells per ml.
Line 155 - why not also use a medium without either a carbon or sulfur source? That would have made the study stronger.
Lines 248 and 254 - Figures 1 and 2 are referred to on these lines, but in each case only one plate of each multiplated figure is relevant to the text at these points of the paper. Figure 1 also has data for TDA, not yet shown in the results - perhaps explain this in text (for example add "see below" for the TDA results). Text on line 254 refers to Figure 2 but is only relevant for Figure 2c (the phylogeny). Figure 2a is later referred to on line 282. Need to cite figure plates in order of appearance in the text. Same thing for citing 2b (line 289). Also on line 289, the rest of the data in Figure 1 are finally discussed but Figure 1 is not cited here. Fix all of this.
Figure 3 - need to label the peaks in the figure.
Line 295 - clarify that it was the extracts, not the coral species, that "exhibited antimicrobial activity".
Line 320 - antimicrobial what?

---

## Round 0.2 · Minor Revisions

The authors have addressed satisfactorily the comments/suggestions from the previous review.

Although the manuscript could be Accepted at this point, Reviewer 2 has provided a couple of suggestions on the revised version which I would encourage the authors to consider. Hence, I am returning it to you with a Minor Revision decision in case you wanted to incorporate any of these final comments.

Reviewer 2 ·

Basic reporting

Basic reporting is fine.

Experimental design

See general comments to author.

Validity of the findings

See general comments to author.

Additional comments

The authors have addressed most of my concerns. I do have a few comments about their response.

1) Response about C/S source: "We are not sure to understand the query. Bacteria require carbon and sulfur to sustain their growth. Incubating a bacterial strain in a medium without one of these essential building blocks will only result in a complete absence of growth." One of the points of the paper is that coral-associated bacteria can metabolize DMSP. Obviously there is a cellular need for both C and S. Growth on media that had no C or S would demonstrate that DMSP would provide a source of BOTH elements, not just one. I’m surprised the author does not understand this simple point.

2) Another response: Line 320 - antimicrobial what? "We are not sure what the reviewer is referring to here. There is no mention of the word antimicrobial line 320."
Here is line 320 copied from the first version of the paper:
319 coral-associated bacteria (Rypien et al. 2010; Shnit-Orland & Kushmaro 2009; Vizcaino et al.
320 2010). Its resistance to antimicrobial is considerably greater than other marine pathogens such as
321 V. parahaemolyticus
Again, need to modify.

3) One of the responses to the other reviewer's concerns about not finding TDA was "Indeed, we only sampled one colony per species, from a location more than 100 km away from the site where the TDA-producing bacteria was isolated and without prior characterization of the bacterial communities present in the colony sampled". This addition is very important, but weakens the study. The TDA analysis should have been performed on the same corals from which the TDA producing isolates were obtained. I would have flagged this also in the first version since it affects the conclusions of the paper. The conclusion now (same paragraph) is that " it would be premature to rule out possible TDA biosynthesis in corals", which is very weak.

---

## Round 0.3 · accepted · Accept

The authors have addressed satisfactorily all the documents from previous reviews.